# Intersectional inequity in knowledge, attitude, and testing related to HIV in Ethiopia: People with multiple disadvantages are left behind

**Aklilu Endalamaw**[1,2]*, **Charles F. Gilks**[1], **Resham B. Khatri**[1], **Yibeltal Assefa**[1]

**1** School of Public Health, The University of Queensland, Brisbane, Australia, **2** College of Medicine and Health Sciences, Bahir Dar University, Bahir Dar, Ethiopia

* yaklilu12@gmail.com

**Data Availability Statement:** All data underlying the findings are provided in the submitted manuscript. The Ethiopian demographic health survey data that we obtained from DHS was

## Abstract

Intersectionality pinpoints intersecting factors that empower or oppress people with multiple (dis)advantageous conditions. This study examined intersectional inequity in knowledge, attitudes, and testing related to HIV among adults aged 15 to 49 years in Ethiopia. This study used nationally representative 2016 Ethiopian Demographic Health Survey data. The sample size was 27,261 for knowledge about HIV/AIDS and 25,542 for attitude towards people living with HIV and HIV testing. Triple (dis)advantage groups were based on wealth status, education status, and residence. The triple advantages variables specifically are urban residents, the educated, and those who belong to households of high wealth status, while the triple disadvantages are rural residents, the uneducated, and those who live in poor household wealth rank. A multilevel logistic regression analysis was employed. Adjusted odds ratios (aOR) and confidence intervals (CI) with a P-value $\leq 0.05$ were considered statistically significant. Based on descriptive analysis, 27.9% (95% CI: 26.5%, 29.3%) of adults had comprehensive knowledge about HIV/AIDS, 39.8% (95% CI: 37.6, 41.9%) exhibited accepting attitude towards people living with HIV, and 20.4% (95% CI: 19.1%, 21.8%) undergo HIV testing. Comprehensive knowledge about HIV/AIDS, accepting attitude towards people living with HIV, and HIV testing was 47.0%, 75.7%, and 36.1% among those with triple advantages, and 13.9%, 16.0% and 8.7% among those with triple non-advantages, respectively. The odds of having comprehensive knowledge about HIV/AIDS, accepting attitude towards people living with HIV, and HIV testing were about three (aOR = 3.4; 95% CI: 2.76 to 4.21), seven (aOR = 7.3; 95% CI: 5.79 to 9.24) and five (aOR = 4.7; 95% CI: 3.60 to 6.10) times higher for triple forms of advantage than triple disadvantages, respectively. The findings of this study imply that Ethiopia will not achieve the proposed targets for HIV/AIDS services unless it prioritises individuals who live under multiple disadvantaged conditions.

## Introduction

Equity analysis using an intersectional lens plays a significant role in the understanding of complex and chronic health issues. For example, HIV/AIDS is a pandemic of intersectional

acquired through an official request made to DHS. To access the DHS Program, please visit their website at https://dhsprogram.com/.

**Funding:** The authors received no specific funding for this work.

inequity, fuelled by gender, income, and education status inequities at the individual, community, and programme levels [1]. Hence, HIV/AIDS prevention and control programmes require a complex start-up with an understanding of intersectionality in providing behavioural services to promote knowledge and attitude, as well as HIV testing [2]. HIV testing is one of the global fast-track goals aimed at reaching 95% of people living with HIV expected to be tested by 2025, which requires contemplating several obstacles [3]. For example, Ethiopia plans that 90% of key and priority populations will have comprehensive knowledge about HIV/AIDS and an accepting attitude towards people living with HIV, as well as knowing their HIV status by 2025 [4]. Ethiopia is among the top fifteen African countries based on the number of people living with HIV (0.62 million) [5], demanding continuous evidence of the comprehensive HIV prevention services.

There have been studies on the overall status and associated factors of knowledge about HIV/AIDS among different population groups in Ethiopia. For instance, a study conducted in 2020 found that 51.4% of university students had knowledge about HIV/AIDS, and this was positively associated with a better education status and a higher income level [6]. Additionally, another study revealed that 72% of adolescent students had a positive attitude towards HIV/AIDS [7]. Attitudes towards people living with HIV manifest as stigma and discrimination. Previous studies investigated stigma and discrimination using similar questions related to attitudes towards people living with HIV [8, 9]. This issue remains a continuous concern in implementing prevention and control strategies for HIV/AIDS [10]. Similarly, a meta-analysis finding based on published article from 2010 to 2019 revealed that the HIV status coverage was about 78% among pregnant women in Ethiopia, as it was much higher among urban residents [11]. These and other available studies have emphasised only one dimension of social categories, such as residence or education status [6, 7, 12–15], while researchers argue the importance of investigating intersectional inequity. Researchers and stakeholders accept the importance of the intersectional approach [16]. The challenge was the lack of evidence on intersectionality in low-income countries, including Ethiopia [17]. This implies that studies on intersectional inequity may contribute to equitable HIV/AIDS services provision.

Intersectionality is a public health concept that plays a critical role in service provision and utilisation as well as disease prevention and control [18]. It is the sharing of characters in overlapping or interdependent social identities, social strata or divisions considered non-medical factors of health [19], such as race, gender, economic power, and class [20]. It can also be seen as an analytical framework for exploring inequity that evolves from intersecting disadvantages [21], arguing that an individual with combined disadvantaged character has a lower probability of accessing services than the single disadvantaged character. Crenshaw described intersectionality as inequities due to individuals being in disadvantageous classes; for instance, black in race and women in sexual orientation [22]. Hurtado also argued how the multiple sources of inequity result in intersectional identities and unveiled an examination of intersectionality inequity [23].

Intersectionality challenges progress towards national and global health targets. Inequity arises from underprivileged social classes in dynamic cultural, economic, political, and health care contexts [24]. For example, people in low socioeconomic conditions experienced lower access to health services, revealing the presence of inequity in healthcare [25]. This inequity is also discernible among individuals with multiple disadvantages [26], again linked to intersectionality theory, involving social and community position, and programme practice [27–29]. Overall, the interaction of social identities is apparent in health services, including HIV/AIDS services [30–32], which needs to be addressed for equality and inclusiveness [33–36] and implies the need for assessing the extent of inequities at the intersection despite known individual determinants [34].

As a result, there has been increasing global attention to intersectionality in research, policies, and programmes. The recent global health target under the Sustainable Development Goals highlights the provision of health services to all people who need them, irrespective of individual backgrounds [37]. The Joint United Nations Programme for HIV/AIDS (UNAIDS) advised assessing intersecting inequities towards ending AIDS epidemics [38]. This underscores the programmatic and policy relevance of investigating intersectionality and inequity in health services [30, 36, 39, 40].

Therefore, we argue individuals with multiple disadvantages will have lower knowledge about HIV/AIDS, an unfavourable attitude towards people living with HIV, and a lower chance of being tested for HIV. The findings will have clinical, programme and policy implications. This study investigated intersectional inequity in knowledge about HIV/AIDS, attitudes towards people living with HIV, and HIV testing in Ethiopia.

## Methods

### Study design and setting

A population-based cross-sectional study design was conducted in Ethiopia (S1 Checklist). Ethiopia is in eastern Africa. Ethiopia is Africa's second-most populated country with a population of nearly 126 million [41], ranked 153$^{rd}$ out of 167 countries in the overall Prosperity Index [42], and ranked 97 out of 156 countries based on the gender gap index score [43]. Around 21.3% population lives in urban areas [44]. This country is geographically divided into ten regions and two cities. These are Tigray, Amhara, Afar, Benshanigul Gumuz, Gambela, Harar, Somali, Oromia, Southern Nations Nationalities and People Region (SNNPR), Sidama, and two cities: Dire Dawa and Addis Ababa. Sidama is a new region that was in SNNPR before 2020. The last Ethiopian-based population-based survey with HIV/AIDS-related indicators was conducted in 2016, in which nine geographic regions were included. The country's HIV/AIDS-related strategic plans and clinical mentoring guidelines are based on the 2016 population-based survey [45].

### Conceptual framework

Using the research questions with a focus on dependent and independent variables, the conceptual framework for the current study was adapted from the social determinants of health concepts. The WHO has developed the conceptual framework for social determinants of health using socioeconomic and political context (governance, microeconomic, public and social policies, cultural and societal values) and socioeconomic position (gender, ethnicity, education, occupation, income) as structural determinants, and material circumstances (living conditions), behavioural and biological factors, and psychosocial factors as intermediary social determinants of health equity and wellbeing as the outcome [46]. Commission of the Pan American Health Organisation has included intersectionality as a social determinant of health equity [47].

The current study has three outcome variables related to inequities: knowledge about HIV/AIDS, an accepting attitude towards people living with HIV, and recent HIV tests across explanatory variables, which are social determinants of health. Fig 1 displays the social determinants adapted from the social determinants of health framework (Fig 1).

### Variables

**Outcome variables.** Comprehensive knowledge about HIV/AIDS, attitude towards people living with HIV, and HIV testing are the outcome variables based on Demographic Health

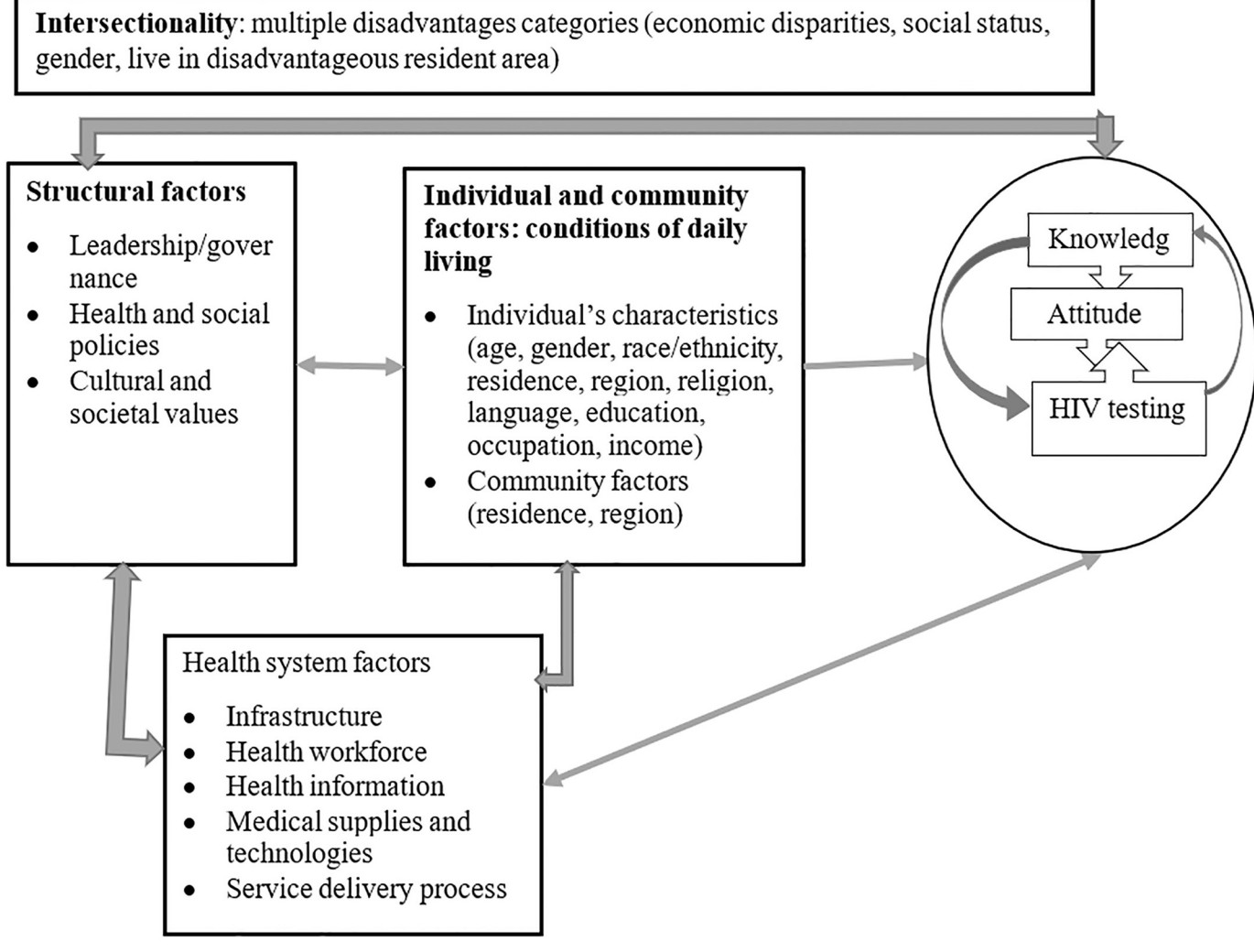

**Fig 1. Conceptual framework adapted based on the social determinants of health framework.**

Survey (DHS) definition [48, 49]. A series of questions were used to generate the level of knowledge. All adults who had ever heard of HIV/AIDS responded to five questions. These are knowing about the two most common methods to prevent HIV/AIDS infection ('consistent condom use' and 'limiting the number of sexual partners to one HIV uninfected faithful partner'), answering correctly to 'a healthy-looking person can have HIV/AIDS' and rejecting the two misconceptions about HIV/AIDS, which are 'a person can get HIV from a mosquito bite', and 'a person can get HIV by sharing a meal with people living with HIV'. Regarding attitude towards people living with HIV, respondents who have heard of HIV/AIDS were asked two consecutive questions: 1) 'would you buy fresh vegetables from a shopkeeper/vendor who had HIV/AIDS?' and 2) 'should children with HIV not be allowed to attend school with children without HIV?' Those who answered two attitude questions positively were considered to have accepting attitude, while those who responded negatively to either of the questions were recorded as having no accepting attitude. The third outcome variable (recent HIV testing and received test result) is whether respondents tested for HIV within 12 months preceding the survey and received test results, represented by HIV testing status in the current manuscript.

**Independent variables.** The current study examined the intersection of education status, household wealth status, and residence as well as employment status and gender towards HIV/AIDS related knowledge, attitude, and testing. This study classified residence, socioeconomic status, and education status as triple intersections while gender and employment status were selected as intersection variables. Of the many distinctions in the social determinants of health, gender and employment status have been historically integrated; women are usually housewives and men are employed [50]. Similarly, residence, education, and economic status are interacting social determinants. These intersecting variables result in health service uptake or coverage disparity between individuals [51]. Hence, looking for the difference between advantaged and disadvantaged groups based on the junction of gender and employment status, and residence area, education, and income status could share several factors. The clustering effect of these intersectional determinants was examined in relation to outcome variables. Before intersection variables were generated, each variable had a dichotomous category. For example, in EDHS, education status originally had four categories (no education, primary education, secondary education, and tertiary education), dichotomised as non-educated and educated (primary, secondary, and tertiary education) in this study. Similarly, wealth status originally had five classes (poorest, poor, medium, rich, and richest), classified as poor (poorer and poorest) and rich (medium, richer, and richest) in the current study. The residence has urban and rural categories. Thus, the categories of the triple intersectional variable are poor uneducated rural (PUR), poor uneducated urban (PUU), poor uneducated rural (PER), poor educated urban (PEU), rich uneducated rural (RUR), rich educated rural (RER), rich uneducated urban (RUU), and rich educated urban (REU). Additionally, the categories of two intersectional variables are unemployed female, unemployed male, employed female, and employed male.

Other exploratory variables involved in the overall analysis of all outcome variables were age category in years (15 to 19, 20 to 24, 25 to 29, 30 to 34, 35 to 39, 40 to 44, 45 to 49), marital status (married, never married, widowed/divorced/no longer living together/separated), religion (orthodox, Catholic, Protestant, Muslim, others), sex of household head (male, female), reading newspaper (no, yes), listening to the radio (no, yes), watching television (no, yes), and region (Tigray, Afar, Amhara, Oromia, Somali, Benshangul-Gumuz, SNNPR, Gambela, Harari, Addis Ababa, Dire Dawa). Region and variable due to the triple (dis)advantageous classes were considered as community-level determinants. Other variables were considered as individual-level determinants. Ever been tested for HIV (no, yes) was included in the analysis of comprehensive knowledge about HIV/AIDS and accepting attitude towards people living with HIV. Comprehensive knowledge about HIV/AIDS (no, yes) was included in the analysis of accepting attitudes and HIV testing. Accepting attitudes towards people living with HIV (no, yes) was included in the analysis of HIV testing.

## Sampling technique and sample size

The 2016 Ethiopian Demographic and Health Survey (EDHS) followed the multistage sampling method, which involves stratification, clustering, and sample selection over two stages [45]. Urban and rural areas were grouped into nine regions and two city administrations as the basis for stratification. Adults who gave consent and agreed to participate in the study responded to the administered questionnaire. A total of 28,371 adults aged from 15 to 59 years were interviewed in EDHS. However, 1,110 adults aged 50 years and above were excluded because current study population was adults 15 to 49 years old. Therefore, the sample size was 27,261 for comprehensive knowledge about HIV/AIDS. Regarding the sample size of accepting attitudes towards people living with HIV, only those who aware of HIV/AIDS who were eligible for attitude-related questions. Based on this, 1,719 adults did not have awareness about

HIV/AIDS and were excluded from the attitudes-related questions. Therefore, 25,542 adults participated to estimate intersectional inequity in accepting attitudes towards people living with HIV. Similarly, the sample size for HIV testing was 25,542 because accepting attitude towards people living with HIV was one of the independent variables in HIV testing. Thus, participants eligible for attitude estimation were automatically eligible for intersectional inequity analysis in HIV testing. The recruitment period for study participants and data collection period was from January to June 2016.

## Data quality control

The EDHS data quality was assured with the provision of training for data collectors, supervisors, and field editors, conducting ongoing supervision, using standardised and translated questionnaires into national and local languages (e.g., Amharic, Oromiffa, Tigrigna) and data processing specialists for data entry and management. A systematic bias was handled throughout this phase [45]. After obtaining data from DHS, proper data management includes appending women's and men's data, handling missed observations through missing completely at random, recoding, and variable recategorisation was properly conducted.

## Statistical analysis

Ethiopian demographic health survey has collected multilevel data at a hierarchical level. It is important to note that we conducted the analysis after checking all the assumption. We checked the Chi-square assumption test, including 'expected value of cells should be 5 or greater in at least 80% of cells'. Multilevel logistic regression is used to analyse multilevel data with binary outcomes [52]. Thus, multilevel logistic regression analysis was run to estimate the effect size of intersectional determinants (uneducated, poor and rural residents versus educated rich and urban resident) and unemployed women versus employed men. Residence, education, wealth status, sex and employment status were excluded from multilevel analysis because intersectional variables were established from these variables. The multicollinearity test provided that the mean-variance inflation factor for variables fitted into the final model was 1.26 (maximum 1.59 and minimum 1.07) for comprehensive knowledge about HIV/AIDS, 1.25 (maximum 1.58 and minimum 1.08) for attitude towards people living with HIV, and 1.25 (maximum 1.59 and minimum 1.08) for recent HIV testing and received test result. Then, we compared intraclass correlation (ICC), Akaike Information Criterion (AIC) and Bayesian Information Criterion (BIC) for the null-model, model-I (individual level determinants), model-II (community-level determinants), and model-III (individual-and community-level factors) (S1 Table). 'The ICC quantifies the proportion of observed variation in the outcomes that is attributable to the effect of clustering' [52]. The current study denotes the variation in the knowledge, attitude, and HIV testing that is because of clustering. In the multilevel logistic regression, region and triple intersectional variables were taken as community-level variables. Finally, standard and multilevel logistic regression models were compared for AIC, BIC and loglikelihood estimates. Generally, the lower AIC and BIC, and the higher loglikelihood estimate was considered as best-fit model [53]. Adjusted odds ratio (aOR) and 95% confidence interval (CI) with a P-value ≤ 0.05 were considered statistically significant.

## Ethics statement

Ethical approval was obtained from DHS (https://dhsprogram.com/). The University of Queensland Institutional Ethical Review Board also exempted the ethical issue of this research (approval project number: 2022/HE001760). Consent and confidentiality were responsibly

managed by the EDHS data collector team during data collection. To illustrate, informed consent was obtained from parents to collect data from children [45].

## Results

### Participants characteristics

Comprehensive knowledge about HIV/AIDS among adults, accepting attitude towards people living with HIV, and HIV testing was 27.9% (95% CI: 26.5%, 29.3%), 39.8% (37.6, 41.9%), and 20.4% (95% CI: 19.1%, 21.8%), respectively. Comprehensive knowledge about HIV/AIDS among unemployed females was 17.6% and 38.9% among employed males. Accepting attitudes among unemployed females was 32.1% and 44.1% among employed males. Recent HIV test among unemployed females was 16.9% and 19.1% among employed males (Table 1).

Comprehensive knowledge about HIV/AIDS among triple disadvantaged adults 15 to 49 years was 13.9% and 47.0% among triple advantaged. Accepting attitudes towards people living with HIV was 16.0% among triple disadvantaged and 75.7% among triple advantaged. HIV testing among triple disadvantaged adults was 7.7% and 35.9% among triple advantaged (Fig 2).

### Intersectional determinants

The ICC in the null model implied that 13.7%, 29.1%, and 21.4% of the total variance in comprehensive knowledge about HIV/AIDS, accepting attitudes towards people living with HIV, and HIV testing, respectively, was credited to community-level determinants. In Model_I only individual-related variables were included. The ICC in Model_I indicated that 9.7%, 17.9%, and 14.0% of the variations of comprehensive knowledge about HIV/AIDS, accepting attitudes towards people living with HIV, and HIV testing, respectively, were accountable to differences across community-level factors. In Model_II, only community-level factors were added. The ICC showed that differences between community-level determinants account 6.1%, 11.0%, and 7.7% of the variation for comprehensive knowledge about HIV/AIDS, accepting attitude towards people living with HIV, and HIV testing, respectively. In Model_III, both individual- and community-level factors were included simultaneously. The ICC values of 6.3%, 10.9%, and 9.2% of the variability in comprehensive knowledge about HIV/AIDS, accepting attitudes towards people living with HIV, and HIV testing, respectively, were accountable to differences between community level factors (S1 Table).

Table 2 shows the multilevel logistic regression results of an association between independents and outcome variables. Relative to adults with triple disadvantage (PUR), the odds of comprehensive knowledge about HIV/AIDS among adults with triple advantage (RUE) were more than four-fold higher (aOR = 3.4; 95% CI: 2.76, 4.21) than their counterparts. Poor, educated, urban resident (PEU) adults had higher odds (aOR = 3.2; 95%CI = 1.39, 7.18) of comprehensive knowledge about HIV/AIDS compared to adults with triple disadvantage. The odds of comprehensive knowledge about HIV/AIDS among rich, educated, rural resident adults (RER) were two-fold higher (aOR = 2.0; 95% CI: 1.74, 2.40) than triple disadvantaged groups. The odds of comprehensive knowledge about HIV/AIDS among poor, educated and rural residents (PER) were about two-fold higher (aOR = 1.8; 95% CI: 1.50, 2.05) than triple disadvantage (PUR).

The odds of accepting attitudes towards people living with HIV among adults with triple advantages REU were about seven-fold higher (aOR = 7.3; 95% CI: 5.79 to 9.24) than those with triple disadvantages (poor, uneducated and rural). Similarly, poor adults with education and residence advantage (PEU) had higher odds (aOR = 2.5; 95% CI = 1.23 to 5.12) of accepting attitude towards people living with HIV compared to adults with triple disadvantage. The

**Table 1. Characteristics of study participants and distribution of outcome variables across them in Ethiopia.**

| Variables | Comprehensive knowledge about HIV/AIDS (n = 27,261) | | Attitude towards People living with HIV (n = 25,542) | | HIV testing (n = 25,542) | |
|---|---|---|---|---|---|---|
| | Participant | Yes (%) | Participant | Yes (%) | Participant | Yes (%) |
| **Age in year** | | | | | | |
| 15–19 | 5,947 | 29.9 | 5,481 | 45.0 | 5,481 | 11.6 |
| 20–24 | 4,640 | 31.3 | 4,414 | 44.8 | 4,414 | 24.9 |
| 25–29 | 4,929 | 28.2 | 4,630 | 41.4 | 4,630 | 27.0 |
| 30–34 | 3,976 | 26.5 | 3,742 | 36.4 | 3,742 | 23.0 |
| 35–39 | 3,314 | 25.9 | 3,095 | 34.5 | 3,095 | 19.4 |
| 40–44 | 2,493 | 23.2 | 23,50 | 34.0 | 2,350 | 19.8 |
| 45–49 | 1,962 | 24.8 | 1,830 | 30.8 | 1,830 | 15.9 |
| **Sex** | | | | | | |
| Men | 11,594 | 38.3 | 11,160 | 45.2 | 11,160 | 19.4 |
| Women | 15,667 | 20.2 | 14,382 | 35.5 | 14,382 | 21.1 |
| **Residence** | | | | | | |
| Urban | 5,774 | 43.08 | 5,624 | 71.0 | 5,624 | 35.4 |
| Rural | 21,487 | 23.79 | 19,918 | 30.9 | 19,918 | 16.1 |
| **Marital status** | | | | | | |
| Never married | 8,909 | 35.6 | 8,384 | 52.0 | 8,384 | 16.3 |
| Married and living together | 16,648 | 24.1 | 15,593 | 32.9 | 15,592 | 22.0 |
| Widowed/divorced/no longer living together/separated | 1,704 | 24.1 | 1,566 | 42.7 | 1,566 | 25.7 |
| **Religion** | | | | | | |
| Orthodox | 11,934 | 33.2 | 11,442 | 48.8 | 11,442 | 25.1 |
| Catholic | 197 | 21.3 | 184 | 31.5 | 184 | 16.3 |
| Protestant | 6,228 | 25.8 | 5,898 | 32.2 | 5,898 | 15.9 |
| Muslim | 8,534 | 22.7 | 7,686 | 33.4 | 7,686 | 17.2 |
| Others | 368 | 15.2 | 332 | 13.0 | 332 | 13.3 |
| **Education status** | | | | | | |
| Non-educated | 10,690 | 15.2 | 9,560 | 22.0 | 9,560 | 14.6 |
| Educated | 16,571 | 36.0 | 15,982 | 50.4 | 15,982 | 23.8 |
| **Household wealth status** | | | | | | |
| Poor | 9,390 | 20.0 | 8,444 | 23.9 | 8,444 | 10.9 |
| Rich | 17,871 | 32.0 | 17,098 | 47.6 | 17,098 | 25.0 |
| **Employment status** | | | | | | |
| Not employed | 8,737 | 19.0 | 7,908 | 35.0 | 7,908 | 17.9 |
| Employed | 18,524 | 32.1 | 17, 634 | 41.9 | 17,634 | 21.5 |
| **Sex of household head** | | | | | | |
| Male | 22,009 | 27.2 | 20,630 | 37.4 | 20,630 | 19.8 |
| Female | 5,252 | 19.3 | 4,912 | 49.5 | 4,912 | 22.9 |
| **Reading newspaper** | | | | | | |
| No | 21,909 | 23.7 | 20,309 | 33.6 | 20,309 | 17.8 |
| Yes | 5,352 | 45.1 | 5,233 | 63.8 | 5,233 | 30.5 |
| **Listening to the radio** | | | | | | |
| No | 16,065 | 22.1 | 14,719 | 32.3 | 14,719 | 15.7 |
| Yes | 11,196 | 36.1 | 10, 823 | 49.8 | 10,823 | 26.8 |
| **Watching television** | | | | | | |
| No | 17,262 | 19.7 | 15,791 | 28.7 | 15,791 | 15.2 |
| Yes | 9,999 | 42.0 | 9,751 | 57.7 | 9,751 | 28.8 |

*(Continued)*

**Table 1.** (Continued)

| Variables | Comprehensive knowledge about HIV/AIDS (n = 27,261) | | Attitude towards People living with HIV (n = 25,542) | | HIV testing (n = 25,542) | |
|---|---|---|---|---|---|---|
| | Participant | Yes (%) | Participant | Yes (%) | Participant | Yes (%) |
| **Region** | | | | | | |
| Tigray | 1,836 | 32.0 | 1,789 | 45.0 | 1,789 | 29.6 |
| Afar | 210 | 20.1 | 194 | 42.1 | 194 | 27.4 |
| Amhara | 6,622 | 31.7 | 6,369 | 46.3 | 6,369 | 22.5 |
| Oromia | 10,099 | 25.1 | 9,227 | 34.1 | 9,227 | 16.3 |
| Somali | 759 | 6.9 | 567 | 17.8 | 567 | 10.8 |
| Benshangul-Gumuz | 278 | 21.2 | 253 | 42.7 | 253 | 25.4 |
| SNNPR | 5,653 | 25.2 | 5,387 | 29.9 | 5,387 | 16.9 |
| Gambela | 78 | 31.2 | 73 | 56.9 | 73 | 37.1 |
| Harari | 67 | 26.4 | 63 | 58.7 | 63 | 23.6 |
| Addis Ababa | 1,502 | 31.3 | 1,474 | 64.4 | 1,474 | 37.1 |
| Dire Dawa | 157 | 46.9 | 146 | 80.4 | 146 | 39.8 |
| **Employment and Gender intersection** | | | | | | |
| Unemployed female | 7,812 | 17.6 | 7,057 | 32.1 | 7,057 | 18.5 |
| Unemployed male | 925 | 30.9 | 851 | 58.6 | 851 | 12.8 |
| Employed female | 7,856 | 22.7 | 7,325 | 38.8 | 7,325 | 23.7 |
| Employed male | 10,668 | 38.9 | 10,309 | 44.1 | 10,309 | 20.0 |
| **Triple intersection** | | | | | | |
| Poor uneducated rural | 5,277 | 13.9 | 4,573 | 16.0 | 4,573 | 7.7 |
| Poor uneducated urban | 98 | 19.4 | 88 | 33.3 | 88 | 19.4 |
| Poor educated rural | 3,892 | 27.9 | 3,663 | 33.5 | 3,663 | 12.8 |
| Poor educated urban | 123 | 32.5 | 120 | 34.2 | 120 | 11.4 |
| Rich uneducated rural | 4,662 | 16.0 | 4,277 | 24.5 | 4,277 | 16.5 |
| Rich educated rural | 7,657 | 33.2 | 7,405 | 42.6 | 7,405 | 20.7 |
| Rich uneducated urban | 653 | 19.1 | 622 | 48.6 | 622 | 34.8 |
| Rich educated urban | 4,899 | 47.0 | 4,794 | 75.7 | 4,794 | 35.9 |
| **Ever been tested for HIV** | | | | | | |
| No | 15,058 | 22.1 | 13,509 | 52.89 | NA | NA |
| Yes | 12,202 | 35.0 | 12,033 | 47.11 | NA | NA |
| **Comprehensive knowledge about HIV/AIDS** | | | | | | |
| No | NA | NA | 18,049 | 31.6 | 18,049 | 18.1 |
| Yes | NA | NA | 7,493 | 59.4 | 7,493 | 25.8 |
| **Accepting attitude towards people living with HIV** | | | | | | |
| No | NA | NA | NA | NA | 15,389 | 15.9 |
| Yes | NA | NA | NA | NA | 10,153 | 27.2 |

NA: not applicable

odds of accepting attitudes among adults with wealth and education advantaged but rural residence (RER) were three- fold higher (aOR = 2.6; 95% CI: 2.11 to 3.11) than triple disadvantage groups.

## Discussion

The findings of this study suggest the presence of intersectional inequity, with significant differences in knowledge about HIV/AIDS, attitude towards people living with HIV, and HIV

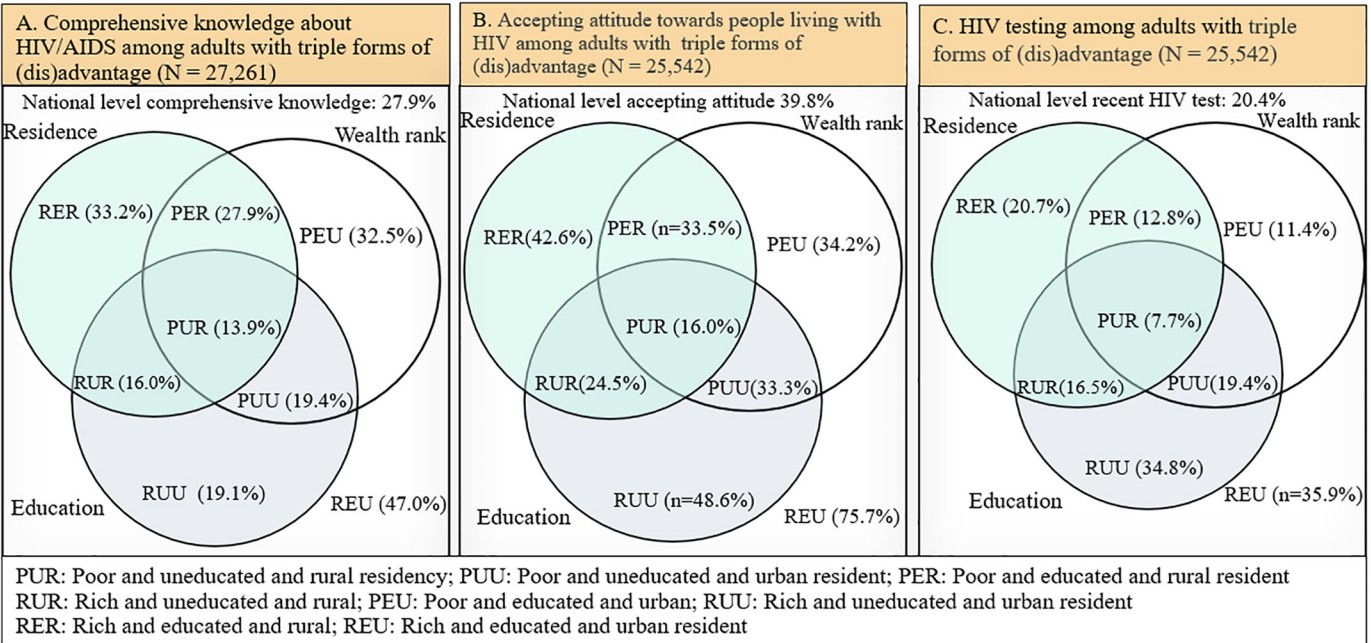

**Fig 2. Comprehensive knowledge, accepting attitude, and HIV testing among adults 15 to 49 years with overlapping (dis)advantages in Ethiopia, 2016.**

testing. Triple-disadvantaged groups have a lower accepting attitudes towards people living with HIV, knowledge about HIV/AIDS, and HIV testing compared to other groups. Economic and demographic determinants cannot solely explain inequity; comprehensive knowledge about HIV/AIDS and having ever been tested for HIV significantly affect attitude towards people living with HIV. Similarly, knowledge and attitude were precursors of recent HIV testing and received test results.

The odds of comprehensive knowledge about HIV/AIDS among unemployed females (double disadvantages) were lower than among employed males (double advantages). In most studies, gender was researched in intersectional inequity with other variables (e.g., race) [54]. Others added that gender influences health and behavioural characteristics [54–56]. It may bring the research and programmes closer to gender-based intersectionality because historical health disparities are recorded in these social categories. In all these cases, individuals with multiple disadvantages lack services and suffer behavioural problems. Although employed males had better knowledge about HIV/AIDS, they had a lower probability of receiving HIV testing than unemployed females. This indicates the flexibility of intersectionality due to males' masculine behaviour. Despite controversial findings in the literature, males might have lower health-seeking behaviour than females. For example, a study in Kenya concluded women had lower health-seeking behaviour than men [57]. The UNAIDS also agreed that HIV/AIDS inequities are attributed to the masculine behaviour of males, who thereby do not use the expected extent of services [58].

Individuals with three non-advantageous social classes (uneducated, poor, and rural residents) also had a lower HIV/AIDS knowledge level and an accepting attitude towards people living with HIV, as well as being less likely to undertake HIV testing. The current study also found that poor rural residents who were educated, rich educated rural residents, and rich uneducated urban residents had better knowledge, attitude, and HIV testing than people with triple disadvantages. This study aligns with another study conducted in maternal care [26].

**Table 2. Multilevel logistic regression analysis of variables to comprehensive knowledge about HIV/AIDS, accepting attitude towards people living with HIV, and HIV testing among adults 15 to 49 years in Ethiopia, 2016.**

| Variables | Comprehensive knowledge about HIV/AIDS aOR (95% CI) | Attitude towards people living with HIV aOR (95% CI) | HIV testing aOR (95% CI) |
|---|---|---|---|
| **Age (reference: 15–19 years)** | | | |
| 20–24 | 1.10 (0.94, 1.28) | 1.27 (1.10, 1.48)** | 2.19 (1.81, 2.64)*** |
| 25–29 | 1.03 (0.88, 1.21) | 1.27 (1.06, 1.52)* | 2.20 (1.76, 2.74)*** |
| 30–34 | 1.15 (0.97, 1.37) | 1.29 (1.07, 1.56)** | 1.84 (1.45, 2.35)*** |
| 35–39 | 1.19 (0.99, 1.44) | 1.26 (1.03, 1.54)* | 1.41 (1.11, 1.79)** |
| 40–44 | 0.91 (0.74, 1.12) | 1.17 (0.95, 1.45) | 1.43 (1.09, 1.88)** |
| 45–49 | 1.10 (0.88, 1.37) | 1.04 (0.82, 1.33) | 1.05 (0.77, 1.43) |
| **Marital status (reference: married)** | | | |
| Never married | 1.21 (1.07, 1.38)** | 1.44 (1.25, 1.66)*** | 0.48 (0.41, 0.56)*** |
| Widowed/divorced/no longer living together/separated | 0.94 (0.79, 1.12) | 1.10 (0.91, 1.34) | 0.86 (0.69, 1.06) |
| **Religion (reference: Orthodox)** | | | |
| Catholic | 0.74 (0.44, 1.25) | 1.08 (0.59, 1.96) | 0.62 (0.34, 1.14) |
| Protestant | 0.91 (0.73, 1.15) | 0.91 (0.74, 1.11) | 0.90 (0.71, 1.13) |
| Muslim | 1.04 (0.91, 1.19) | 0.91 (0.76, 1.09) | 1.11 (0.90, 1.38) |
| Others | 0.74 (0.41, 1.33) | 0.39 (0.17, 0.88)* | 1.12 (0.66, 1.92) |
| **Sex of household head (reference: male)** | | | |
| Female | 1.13 (1.003, 1.27)* | 1.16 (1.03, 1.31)* | 1.07 (0.94, 1.22) |
| **Reading newspapers (reference: No)** | | | |
| Yes | 1.23 (1.10, 1.34)*** | 1.62 (1.41, 1.85)*** | 1.26 (1.11, 1.43)*** |
| **Listening to the radio (reference: No)** | | | |
| Yes | 1.03 (0.92, 1.15) | 1.06 (0.89, 1.14) | 1.34 (1.19, 1.51)*** |
| **Watching television (reference: No)** | | | |
| Yes | 1.45 (1.27, 1.67)*** | 1.26 (1.11, 1.44)*** | 1.10, 0.96, 1.27) |
| **Gender & employment intersection (Reference: unemployed female)** | | | |
| Unemployed male | 1.39 (1.08, 1.78)* | 1.72 (1.34, 2.22)*** | 0.64 (0.45, 0.91)* |
| Employed female | 1.12 (0.98, 1.28) | 1.01 (0.86, 1.19) | 0.97 (0.85, 1.11) |
| Employed male | 2.43 (2.07, 2.85)*** | 1.25 (0.99, 1.58) | 0.85 (0.73, 0.98)* |
| **Region (reference: Addis Ababa)** | | | |
| Tigray | 1.25 (1.01, 1.55)* | 0.76 (0.57, 1.04) | 1.68 (1.34, 2.12)*** |
| Afar | 0.75 (0.55, 1.03) | 0.99 (0.71, 1.39) | 1.52 (1.13, 2.04)** |
| Amhara | 1.35 (1.10, 1.65)** | 1.00 (0.77, 1.31) | 1.16 (0.94, 1.44) |
| Oromia | 0.99 (0.80, 1.23) | 0.55 (0.40, 0.76)*** | 0.59 (0.45, 0.78)*** |
| Somali | 0.28 (0.20, 0.37)*** | 0.29 (0.21, 0.40)*** | 0.52 (0.37, 0.74)*** |
| Benshangul-Gumuz | 0.76 (0.60, 0.97)* | 0.92 (0.68, 1.24) | 1.32 (1.004, 1.76)* |
| SNNPR | 1.08 (0.84, 1.38) | 0.50 (0.38, 0.68)*** | 0.88 (0.67, 1.14) |
| Gambela | 0.92 (0.73, 1.16) | 0.96 (0.69, 1.32) | 1.87 (1.45, 2.41)*** |
| Harari | 0.66 (0.49, 0.89)** | 0.97 (0.74, 1.27) | 0.68 (0.53, 0.87)** |

(*Continued*)

**Table 2.** (Continued)

| Variables | Comprehensive knowledge about HIV/AIDS aOR (95% CI) | Attitude towards people living with HIV aOR (95% CI) | HIV testing aOR (95% CI) |
|---|---|---|---|
| Dire Dawa | 0.66 (0.49, 0.90)* | 0.96 (0.74, 1.23) | 1.58 (1.27, 1.96)*** |
| **Triple intersection based on wealth index, education, residence status** (reference: Poor uneducated rural (PUR)) | | | |
| Poor uneducated urban (PUU) | 1.64 (0.83, 3.27) | 1.65 (0.70, 3.87) | 3.34 (1.01, 11.12)* |
| Poor educated rural (PER) | 1.75 (1.50, 2.05)*** | 2.07 (1.68, 2.54)*** | 1.80 (1.45, 2.22)*** |
| Poor educated urban (PEU) | 3.16 (1.39, 7.18)** | 2.51 (1.23, 5.12)*** | 2.22 (0.86, 5.78) |
| Rich uneducated rural (RUR) | 1.00 (0.83, 1.21) | 1.35 (1.12, 1.64)** | 1.86 (1.51, 2.30)*** |
| Rich educated rural (RER) | 2.04 (1.74, 2.40)*** | 2.56 (2.11, 3.11)*** | 2.67 (2.15, 3.31)*** |
| Rich uneducated urban (RUU) | 1.27 (0.90, 1.79) | 3.60 (2.66, 4.88)*** | 4.84 (3.59, 6.54)*** |
| Rich educated urban (REU) | 3.41 (2.76, 4.21)*** | 7.3 (5.79, 9.24)*** | 4.69 (3.60, 6.10)*** |
| **Ever been tested for HIV** (reference: No) | | | |
| Yes | 1.38 (1.25, 1.55)*** | 1.34 (1.21, 1.49)*** | _ |
| **Comprehensive knowledge about HIV/AIDS** (Reference: No) | | | |
| Yes | _ | 2.06 (1.85, 2.31)*** | 1.17 (1.04, 1.30)** |
| **Accepting attitude towards people living with HIV** (reference: No) | | | |
| Yes | _ | _ | 1.18 (1.05, 1.31)** |

* p-value≤0.05;

** p-value≤0.01;

*** p-value≤0.00;

SNNPR: Southern Nations, Nationalities and Peoples' Region

Maternal care and HIV/AIDS services are indeed distinct arenas. Still, the social categories for the intersectionality approach have followed similar methods and analyses. Women with triple disadvantages were less likely to attend antenatal and postnatal care [26], similar to the current study. There could be a higher probability that an individual with triple better living conditions (urban resident, educated and rich) has higher intersecting opportunities in living conditions, accessing health information, behavioural change services, and clinical interventions. This is because each advantageous living character's combined opportunities, resources, and power greatly improved behaviour, service coverage, and health outcomes. Urban, better-educated, and individuals in the higher economic class are usually near healthcare because they reside around healthcare settings, have better knowledge about health problems, and can cover direct and indirect healthcare expenditures [59]. In the Ethiopian context, the more educated reside in urban areas and have a higher income, which reflects the interrelationship between each category [60, 61].

Congruent with existing research, the current findings are also in agreement with how HIV/AIDS was assessed at the intersection of race or ethnicity (Latina) and culture or language (monolingual) [12]. In both studies, those with the intersection of disadvantageous identities had a lower level of HIV/AIDS knowledge as compared to their counterparts. Additionally, Ghasemi et al. found that those with triple disadvantages were more likely to experience HIV-

related stigma in Iran [31]. In the current and previous studies, intersectional inequity was profound despite different intersectional determinants. This implies that intersectionality works in different healthcare contexts, revealing that the more disadvantaged social identity has the least service accessibility.

Improving the living conditions of communities is essential, especially by addressing multi-dimensional poverty, enhancing education, and tackling inequalities. Low-income adults face various disadvantages across multiple dimensions, with poverty significantly intersecting with health, education, and overall living conditions. The United Nations Development Programme (UNDP) emphasizes the need for countries to prioritize improving living conditions to pro-mote a healthier population. Notably, a report covering 110 countries reveals that 25 of them successfully halved multidimensional poverty within 15 years. Despite this progress, 17.0% (1.1 billion out of 6.1 billion people) still live in poverty. Of these 1.1 billion individuals, 534 million reside in sub-Saharan Africa [62]. The UNDP further underscores the importance of examining why people are left behind, empowering them, and implementing policies and reforms to address the drivers of poverty [63].

The current finding marks a turning point in embracing intersectionality in policy and clin-ical services. Therefore, the recent evidence suggests a need for a more nuanced emphasis on people with multiple disadvantages. Intersectional inequity should be emphasised in policy formulation and implementation, the health service system, the research environment, and health academics towards knowledge, attitude, and testing in the HIV/AIDS continuum. To integrate this finding into programmes and practises, emphasis should be given first to triple-disadvantaged groups, followed by double- and single-disadvantaged groups. This means interventions to narrow disparities may not be uniform for urban and rural populations, which is important for international and national initiatives. For example, UNAIDS intro-duced 'education plus' initiatives to be implemented from 2021 to 2025, which focus on ado-lescent girls and young women to achieve gender equality in preventing HIV [64]. Thus, implementing this initiative can consider girls who live in lower-income households, lower grade levels, and rural areas (one girl has these three conditions at a time) in the first round of services. The interventions aimed at enhancing knowledge about HIV/AIDS, ending stigma and discrimination, and expanding HIV testing coverage are also essential in addressing inter-sectional inequity. It is essential to adopt culturally sensitive and contextually relevant inter-ventions. These might involve training initiatives and research activities, especially in low- and middle-income countries [10]. The UNDP highlights intersectional determinants, including geography, discrimination, poverty, race, and socioeconomic status. However, some of these factors were not addressed in the current study. Therefore, it is crucial for future researchers to explore intersectionality from multiple dimensions. For instance, examining the intersection of female widows, sex workers, daily labourers, factory workers, long-distance travellers, and other vulnerable groups in the context of fighting HIV/AIDS is essential, as these individuals face a high risk of HIV infection [65].

## Strengths and limitations

The current study is the first of its kind in HIV/AIDS services, and the intersectional subgroup variables had a sufficient sample size, which is likely to minimise random error. Hence, it solved the literature gap in low-income countries, particularly in Ethiopia, because the scarcity of intersectionality evidence challenged its implementation in low-income countries [66].

As to the limitations, knowledge, attitude, and HIV testing were measured based on respon-dents' self-report, which is prone to recall bias. Second, a cross-sectional study design has a 'chicken-egg' dilemma scenario; thus, the current findings cannot represent a cause-and-effect

relationship [67]. Third, some structural and health system determinants were not considered in the analysis because EDHS has not included governance and leadership, policies, and health systems' variables about HIV/AIDS indicators. Fourth, the UNAIDS has implemented a combination HIV prevention program, which encompasses behavioural interventions (e.g., counselling and education program), biomedical services (e.g., HIV testing), and structural interventions (e.g., initiatives to reduce stigma and discrimination) [2]. We have tried to incorporate these three programmes in a manner that allows for measurable evaluation. Inequities in health services may reflects an unfair distribution of resources. Hence, in our study, careful consideration may be required to understand the inequity in accepting attitudes towards people living with HIV and knowledge about HIV/AIDS because these are not services but rather outcomes of certain interventions. An intervention aimed at addressing stigma and discrimination entails fostering accepting attitudes towards people living with HIV. This implies that an individual with lower accepting attitudes may require structural intervention to reduce stigma and discrimination. Similarly, a lower knowledge about HIV/AIDS may reflect a gap in the counselling and education interventions.

## Conclusions

Triple-disadvantaged groups have a lower accepting attitude towards people living with HIV, comprehensive knowledge about HIV/AIDS, and recent HIV testing. Using intersectionality as an equity lens will allow reaching populations belonging to multiple disadvantages. If the intersectional inequity concept is included in the health policy statement, the triple disadvantaged (rural and uneducated and poor) will receive equity-based health care intervention first, followed by the double disadvantaged (e.g., rural and poor and educated), and finally the single disadvantaged (as usual).

It is necessary to design a strategy for achieving equity among people with multiple disadvantages. A combination of individual and public health measures should be delivered with wider attempts to address multiple forms of inequity. Additionally, prioritise intersectionality and inequity as a new agenda to consider at clinical, research, and policy levels sustainably. To understand intersectionality at the macro- and policy-level, a critical HIV/AIDS policy evaluation is required.

## Supporting information

**S1 Table. Intraclass correlation (ICC), Akaike Information Criterion (AIC) and Bayesian Information Criterion (BIC) for the null-model, model-I (individual level determinants), model-II (community-level determinants), and model-III (individual-and community-level factors).**
(PDF)

**S1 Checklist. STROBE checklist: Research reporting checklist for cross-sectional studies.**
(DOCX)

## Author Contributions

**Conceptualization:** Aklilu Endalamaw.

**Data curation:** Aklilu Endalamaw.

**Formal analysis:** Aklilu Endalamaw.

**Investigation:** Aklilu Endalamaw.

**Methodology:** Aklilu Endalamaw.

**Project administration:** Aklilu Endalamaw.

**Software:** Aklilu Endalamaw.

**Supervision:** Charles F. Gilks, Yibeltal Assefa.

**Validation:** Aklilu Endalamaw, Resham B. Khatri, Yibeltal Assefa.

**Visualization:** Aklilu Endalamaw, Charles F. Gilks, Yibeltal Assefa.

**Writing – original draft:** Aklilu Endalamaw.

**Writing – review & editing:** Aklilu Endalamaw, Charles F. Gilks, Resham B. Khatri, Yibeltal Assefa.

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
