## [Decision Letter · Decision Letter 0]

20 Jun 2024

PGPH-D-24-00849

Intersectional inequity in knowledge, attitude, and testing related to HIV in Ethiopia: People with multiple disadvantages are left behind

Dear Dr. Endalamaw,

Thank you for submitting your manuscript to PLOS Global Public Health. After careful consideration, we feel that it has merit but does not fully meet PLOS Global Public Health’s publication criteria as it currently stands. Therefore, we invite you to submit a revised version of the manuscript that addresses the points raised during the review process.

We look forward to receiving your revised manuscript.

Kind regards,

Rayner Kay Jin Tan

Academic Editor

Journal Requirements:

Additional Editor Comments (if provided):

The reviewers have provided some constructive feedback to strengthen the narrative of your manuscript. One suggestion revolves about additional analysis which may be helpful to consider as well, if possible. If such data are not available, do justify the approach in your revision.

Reviewers' comments:

Reviewer's Responses to Questions

**Comments to the Author**

1. Does this manuscript meet PLOS Global Public Health’s publication criteria? Is the manuscript technically sound, and do the data support the conclusions? The manuscript must describe methodologically and ethically rigorous research with conclusions that are appropriately drawn based on the data presented.

Reviewer #1: Yes

Reviewer #2: Yes

2. Has the statistical analysis been performed appropriately and rigorously?

Reviewer #1: Yes

Reviewer #2: Yes

3. Have the authors made all data underlying the findings in their manuscript fully available (please refer to the Data Availability Statement at the start of the manuscript PDF file)?

Reviewer #1: Yes

Reviewer #2: Yes

4. Is the manuscript presented in an intelligible fashion and written in standard English?

Reviewer #1: Yes

Reviewer #2: Yes

5. Review Comments to the Author

Reviewer #1: This is a very good paper that documents people who have not accessed different HIV prevention and control messages and HIV testing services.

I have two major comments:

• The global direction for HIV prevention, control, and treatment has shifted to key and priority populations due to resource constraints. This paper, however, covers the general population. Can you add one variable lumping key and priority populations together and compare it with the rest of the population with regard to the ‘triple disadvantages’? You may not find all key and priority populations in the DHS data but you can use whatever is available like divorced, widowed, separated, military/uniformed people, waitresses (may be as a proxy for female sex workers), daily laborers, factory workers, university students, etc. Such analysis may highlight the gap for policy makers.

• You used the term ‘Marginalisation based on wealth index, education, residence status’. Marginalization is a very strong word to use in this context since no one is deliberately going out of their way to discriminate against them so they don’t access these services. They have just not been served for various reasons.

Minor comment:

• Since you are discussing about inequality, it would have been easier to calculate and interpret odds ratio using rich, urban, educated as reference.

Reviewer #2: A well written manuscript on an important topic. The authors have used a sound methodological approach to investigate the intersectional inequality issues in relation to HIV in Ethiopia. I have a few minor comments that the authors should consider to make it a stronger narrative.

The authors need to add some more narrative about stigma and anti-stigma approach in the introduction and some reflection about this in the discussion sections. Stigma intensifies existing power inequalities and social inequities, profoundly affecting various identities and health outcomes, especially in low and middle-income nations. You can read and include a recently published article on this topic by Majeed et al: Anti-stigma interventions in low-income and middle-income countries: a systematic review: https://pubmed.ncbi.nlm.nih.gov/38707913/

In the discussion section, I feel that authors should reflect on the triple disadvantages and intersectionality and their close association with UNDP's multidimentional poverty index (https://hdr.undp.org/content/2023-global-multidimensional-poverty-index-mpi#/indicies/MPI) or other indices of inequality such as the Human Development Index.

6. PLOS authors have the option to publish the peer review history of their article (what does this mean?). If published, this will include your full peer review and any attached files.

**Do you want your identity to be public for this peer review?** For information about this choice, including consent withdrawal, please see our Privacy Policy.

Reviewer #1: **Yes: **Kesetebirhan D Yirdaw

Reviewer #2: No

---

## [Decision Letter · Decision Letter 1]

31 Jul 2024

Intersectional inequity in knowledge, attitude, and testing related to HIV in Ethiopia: People with multiple disadvantages are left behind

PGPH-D-24-00849R1

Dear Mr. Endalamaw,

We are pleased to inform you that your manuscript 'Intersectional inequity in knowledge, attitude, and testing related to HIV in Ethiopia: People with multiple disadvantages are left behind' has been provisionally accepted for publication in PLOS Global Public Health.

We note that one or more reviewers has recommended that you cite specific previously published works in an earlier round of revision. As always, we recommend that you please review and evaluate the requested works to determine whether they are relevant and should be cited. It is not a requirement to cite these works and you may remove them before the manuscript proceeds to publication. We appreciate your attention to this request.

Best regards,

Julia Robinson

Executive Editor

Reviewer Comments (if any, and for reference):

Reviewer's Responses to Questions

**Comments to the Author**

1. If the authors have adequately addressed your comments raised in a previous round of review and you feel that this manuscript is now acceptable for publication, you may indicate that here to bypass the “Comments to the Author” section, enter your conflict of interest statement in the “Confidential to Editor” section, and submit your "Accept" recommendation.

Reviewer #1: All comments have been addressed

2. Does this manuscript meet PLOS Global Public Health’s publication criteria? Is the manuscript technically sound, and do the data support the conclusions? The manuscript must describe methodologically and ethically rigorous research with conclusions that are appropriately drawn based on the data presented.

Reviewer #1: Yes

3. Has the statistical analysis been performed appropriately and rigorously?

Reviewer #1: Yes

4. Have the authors made all data underlying the findings in their manuscript fully available (please refer to the Data Availability Statement at the start of the manuscript PDF file)?

Reviewer #1: Yes

5. Is the manuscript presented in an intelligible fashion and written in standard English?

Reviewer #1: Yes

6. Review Comments to the Author

Reviewer #1: Comments addressed.

7. PLOS authors have the option to publish the peer review history of their article (what does this mean?). If published, this will include your full peer review and any attached files.

**Do you want your identity to be public for this peer review?** For information about this choice, including consent withdrawal, please see our Privacy Policy.

Reviewer #1: **Yes: **Kesetebirhan Delele Yirdaw
